# Immunomodulatory Role of Thioredoxin Interacting Protein in Cancer’s Impediments: Current Understanding and Therapeutic Implications

**DOI:** 10.3390/vaccines10111902

**Published:** 2022-11-10

**Authors:** Ramkumar Katturajan, Sangeetha Nithiyanandam, Manisha Parthasarathy, Abilash Valsala Gopalakrishnan, Ezhaveni Sathiyamoorthi, Jintae Lee, Thiyagarajan Ramesh, Mahalaxmi Iyer, Sabina Evan Prince, Raja Ganesan

**Affiliations:** 1Department of Biomedical Sciences, School of Biosciences and Technology, Vellore Institute of Technology (VIT), Vellore 632014, Tamil Nadu, India; 2School of Chemical Engineering, Yeungnam University, Gyeongsan 38541, Korea; 3Department of Basic Medical Sciences, College of Medicine, Prince Sattam bin Abdulaziz University, P.O. Box 173, Al-Kharj 11942, Saudi Arabia; 4Livestock Farming and Bioresource Technology, Coimbatore 641003, Tamil Nadu, India; 5Institute for Liver and Digestive Disease, College of Medicine, Hallym University, Chuncheon 24253, Korea

**Keywords:** cancer, immunomodulation, immune cells, TXNIP, JAK-STAT, AMPK, PI3K/Akt

## Abstract

Cancer, which killed ten million people in 2020, is expected to become the world’s leading health problem and financial burden. Despite the development of effective therapeutic approaches, cancer-related deaths have increased by 25.4% in the last ten years. Current therapies promote apoptosis and oxidative stress DNA damage and inhibit inflammatory mediators and angiogenesis from providing temporary relief. Thioredoxin-binding protein (TXNIP) causes oxidative stress by inhibiting the function of the thioredoxin system. It is an important regulator of many redox-related signal transduction pathways in cells. In cancer cells, it functions as a tumor suppressor protein that inhibits cell proliferation. In addition, TXNIP levels in hemocytes increased after immune stimulation, suggesting that TXNIP plays an important role in immunity. Several studies have provided experimental evidence for the immune modulatory role of TXNIP in cancer impediments. TXNIP also has the potential to act against immune cells in cancer by mediating the JAK-STAT, MAPK, and PI3K/Akt pathways. To date, therapies targeting TXNIP in cancer are still under investigation. This review highlights the role of TXNIP in preventing cancer, as well as recent reports describing its functions in various immune cells, signaling pathways, and promoting action against cancer.

## 1. Introduction

Globally, cancer is one of the leading causes of death. By 2050, it will overtake all other diseases combined in terms of total mortality, especially in developing low- and middle-income countries [1]. Malignant tumor treatment is available in numerous forms, including surgical procedures, chemotherapy, immunotherapy, and radiotherapy [2]. Cancer is still a fatal disease on a global scale, as evidenced by the rapid rise in cancer mortality and incidence in many countries. According to the International Agency for Research on Cancer (IARC), there will be approximately 19.3 million new cancer cases and nearly 10.0 million cancer deaths in 2020. By 2040, there will likely be 16.3 million deaths and 30.2 million new cases. Therefore, understanding how genes are regulated is critical if researchers develop new cancer treatment strategies and effective new medications [3].

Thioredoxin-interacting protein (TXNIP), also known as vitamin D upregulated protein-1 (VDUP-1) or thioredoxin-binding protein-2 (TBP-2), is a multifunctional protein that is essential for many cellular processes such as metabolic activities, growth, division, and cell death [4]. The regulation of redox homeostasis is greatly influenced by the TRX/TXNIP system. TXNIP is thought to play a role in tumor suppression by impeding cellular glucose uptake and increasing oxidative stress. Significantly reduced TXNIP expression has been observed in numerous tumors, suggesting its function as a tumor suppressor. TXNIP has been hypothesized as a possible tumor suppressor in cancer due to its capacity to cause oxidative stress-mediated apoptosis in cancer cell lines and tumor tissues. The immune system’s ability to recognize and destroy tumor cells may be controlled by the immunological modulatory properties of TXNIP [5]. As a result, a moderator specifically targeting TXNIP is being developed for cancer treatment. TXNIP has been predicted to control and introduce various immune cells and signaling pathways in cancer. TXNIP has also been mentioned as a potential key factor in cell development in recent findings [6,7]. This evidence suggests that TXNIP plays an important role in cancer and is required for targeted therapy. To the best of our knowledge, this is the first report to be distributed that clearly outlines the role of TXNIP in cancer signaling. In this review, we discuss the implications of the immune modulatory function of TXNIP in various cancers and its role in mediating cellular damage in relation to the dependent cascade.

## 2. TXNIP

Thioredoxin-interacting protein (TXNIP), also known as vitamin D upregulated protein-1 (VDUP-1) or thioredoxin-binding protein-2 (TBP-2), is a multifunctional protein that is essential for many cellular processes such as metabolic activities, growth, division, and cell death [4]. It was initially believed that TXNIP was restricted only to the cytoplasm. However, multiple investigations uncovered the possibility of TXNIP migrating to various intracellular sites in response to oxidative stress. Furthermore, the biological function of TXNIP may be influenced by its internal locations, such as mitochondria and the cell’s surface. Research has also revealed that TXNIP is localized in the plasma membrane [8]. The fundamental function of the human TXNIP, an alpha-arrestin protein with a 46 kDa molecular weight and 391 amino acids in its core and arrestin-like N- and C-termini, is to hinder binding proteins from performing their biological functions [9]. Firstly, TXNIP plays a critical role in selectively inhibiting glucose transporter-1 at the plasma membrane, which has the potential to regulate cellular glucose metabolism and cell division by limiting the propensity of cells to take up glucose [10]. Secondly, through the induction of G1 cell cycle arrest, TXNIP can also impede cell cycle progression [11]. Thirdly, cancer etiology and the efficacy of anticancer therapies are associated with oxidative stress, which is prompted by the aggregation of excessive reactive oxygen species. Thioredoxin is one of the detoxification enzymes of phase II, which, together with catalase, glutathione peroxidases, peroxiredoxins, superoxide dismutase, and glutaredoxin, form the cellular antioxidant defense system [12]. TXNIP also functions as a down-regulator of the thioredoxin mechanism by preventing thioredoxin-1 (TRX-1) from performing its antioxidant action by binding TXNIP to its active cysteine residue, resulting in the upsurge of oxidative stress [13].

Meanwhile, the regulation of redox homeostasis is greatly influenced by the TRX/TXNIP system. Therefore, TXNIP is thought to play a role in tumor suppression by impeding cellular glucose uptake, disrupting the cell cycle, and increasing oxidative stress. Significantly reduced TXNIP expression has been observed in numerous tumors, suggesting its function as a tumor suppressor [14]. In addition, several consecutive studies have shown that cancer patients with elevated TXNIP expression have a better prognosis than those with reduced TXNIP expression, making TXNIP a potential therapeutic target for cancer treatment.

## 3. The Action of TXNIP on Immune Cells

TXNIP controls the TRX system’s function negatively to prevent an excessive tilt of the cellular redox balance toward the reducing side [15]. As a result, the modulation of the immune system is significantly influenced by the TRX system, which regulates the redox equilibrium of cells [16]. Therefore, it is unsurprising that TXNIP may play a putative role in immunity. It has been progressively shown that TXNIP is important in the interaction between oxidative stress and innate host defense mechanisms.

### 3.1. T Helper 17 Cells

T cells are a key component of cell-mediated immunity and a significant subset of adaptive immunity [17]. Until they detect a particular antigen or major histocompatibility complex, T lymphocytes circulate throughout the body in a quiescent, naive condition [18]. Co-stimulatory signals are indispensable for T cells to be effectively activated in addition to antigen-driven signals. Members of the tumor necrosis factor superfamily are the receptors with which T cells can interact to receive co-stimulatory signals [19]. TRX, a multifunctional regulatory protein, is crucial for effective T cell activation and stimulating rapid cell proliferation. High glucose absorption is crucial for the proliferation and upregulation of active T cells since it supplies the building blocks and the energy required [20]. TXNIP, which can bind TRX via intermolecular disulfide bonds, modulates T cell responses, suppresses redox function, and inhibits glucose metabolism; therefore, glucose deficiency may prevent cells from progressing through the G1 phase of the cell cycle, preventing T cells from proliferating and surviving [21]. Naive human T cells exhibit high TXNIP production, whereas T cells stimulated by the T cell receptor exhibit low TXNIP production [22]. T cells that are CD4+ and CD8+ make up the majority of T lymphocytes. An essential factor in modulating the immune response is the production of specific cytokines by activated and differentiated CD4+T cells into diverse effector subtypes [23]. T helper 17 cells (Th17), a lineage of CD4+T helper cells, have become a key target for disease management.

A crucial mechanism that supports T cell activation and differentiation is the “Warburg effect”, which was initially used to explain phenomena in which most cancer cells rely on aerobic glycolysis for their proliferation [24]. These Th17 cells that rely on aerobic glycolysis upregulate glucose transporter-1 (GLUT1) to facilitate sustained cell proliferation. However, this function is inhibited when TXNIP is significantly upregulated under oxidative stress. In addition, the transcription factor HIF (Hypoxia-inducible factor)-1α, which promotes the glycolytic activity of Th17 cells, is critical for Th17 cell development [25]. Another possibility is that TXNIP disrupts the transcription factor HIF-1α, which controls the expression of most glycolytic target genes that regulate proliferation [26]. Therefore, it is plausible that TXNIP either directly or indirectly controls glucose transporter expression. Thus, TXNIP has become a formidable competitor in controlling cell metabolism and proliferation (Figure 1).

### 3.2. Follicular T Helper and B Cells

Antigen activates a subpopulation of naive T cells that migrate to the follicles, where they develop into TFH cells that communicate with and control follicular B cells [27]. The primary T cell lineage that supports B cells’ maturation and proliferation to create antibodies for controlling the humoral immune system’s response is known as follicular T helper cells (TFH) [28]. TFH is also necessary for establishing germinal centers, affinity maturation, the majority of high-specificity antibodies, and memory B cell development. In addition, TFH elicits cytokines such as interleukin (IL)-10 and -21, which stimulate B cell survival and antibody production [29]. In a manner similar to T cells, activated B cells enhance amino acid absorption to maintain metabolic changes and retain cellular redox homeostasis [30]. Antigen-dependent T-B cell interaction is an important first step in establishing efficient B cell immunity. Follicular B2 cells can proliferate without the TRX system by using the glutathione/glutaredoxin pathway to synthesize deoxynucleoside triphosphate (dNTP), although this pathway is less efficient at maintaining proliferation [31].

On the other hand, TXNIP functions as a metabolic barrier in B-cell acute lymphoblastic leukemia and is integral to the progression and functionality of natural killer cells [32]. A further indication is that the redox mechanism of innate-like B lymphocytes differs from that of follicular B2 cells, which is evidence that TRX1 is mostly dispensable in these cells [31]. B cell development and responses depend on the transcription factor paired box protein (PAX)-5, which regulates the expression of TXNIP and directly represses the glucose transporter [33]. Various triggers, such as dietary stimuli, insulin, and glucose, can activate TXNIP to overexpress. When TXNIP is rapidly upregulated, the TRX system is inhibited, and glucose absorption is downregulated, which culminates in a redox imbalance [34].

Moreover, the energy restriction to the pentose phosphate pathway caused by the inhibition of aerobic glycolysis disrupts the production of NADPH and dNTPs [35]. The upregulation of TRX1 system components causes poor clinical outcomes in various tumors, including B-cell malignancies and T-cell acute lymphoblastic leukemia. In clinical settings, chemotherapeutic medications targeting TRX reductase (TRXR) are theorized to function by triggering oxidative stress and death in tumor cells [36]. Muri et al. reported that the negative regulator TXNIP restrains T- and B-cell proliferation during immune responses but is dispensable for their homeostatic maintenance [37]. These findings collectively indicate that TXNIP activation may directly impact follicular T helper and B cells, as illustrated in Figure 1.

### 3.3. Monocytes and Macrophages

Monocytes and macrophages comprise the mononuclear phagocyte system, the cornerstone of innate immunity [38]. The proinflammatory mediator macrophage migration inhibitory factor (MIF) is an essential component of the innate immune response. MIF was first discovered to be a soluble cytokine produced by T cells stimulated during a delayed-type hypersensitivity reaction [39]. After being stimulated by inflammatory mediators or stress, this protein is expressed in various cell types, including macrophages, monocytes, T cells, eosinophils, and endothelial and epithelial cells. MIF expression has been reported to be significantly elevated in many forms of cancer, including prostate, breast, lung, colon, and malignant melanoma [40].

In addition, it stimulates tumor angiogenesis and modulates immunological responses to promote cancer growth. Intracellular MIF induces nuclear factor kappa B (NF-κB) activity [41]. Controlling the survival and death of cancer cells depends heavily on the NF-κB pathway. According to several studies, carcinoma of the breast and lungs, lymphoma, and leukemia cell cultures have persistently active NF-κB, which regulates the growth and spread of cancer [42]. Furthermore, in glioblastoma and ovarian cancer, NF-κB was present at a markedly elevated level; this level has been linked to a negative prognosis [43]. It has been demonstrated that inhibiting NF-κB signaling improves anticancer responses. TXNIP, a well-known inhibitor of NF-κB activity, interacts directly with MIF and may significantly reduce NF-κB activation [44]. Bridging the links between TXNIP and the proteins histone deacetylases (HDAC) and p65 can prevent them from deactivating the growth-suppressive gene, as these proteins are responsible for bolstering cancer cell proliferation [41]. Thus, HDAC/p65 inhibition could be deployed to halt the development of cancer cells.

Furthermore, cytological experiments confirmed that TXNIP promotes monocyte adhesion and motility. According to Rong et al.’s findings, enhanced TXNIP activated the nucleotide oligomerization domain (NOD) like receptor (NLR) family pyrin domain containing (NLRP)3 inflammasome, which triggered a cascade of further inflammatory responses that exacerbated monocyte activation and inflammatory processes [45]. TXNIP was paramount for NLRP3 inflammasome activation and functioned as a gauge for regulating redox signaling molecule concentrations [46]. Targeting the NLRP3 inflammasome and its downstream pathways, independently or in conjunction with chemotherapy and other immunotherapy approaches, could remain a lucrative prospect in managing cancer [47]. Taken together, a deeper insight into HDAC/P65 inhibition and systemic monocyte reprogramming in cancer may contribute to novel therapeutic interventions that have the potential to set long-lasting anticancer immunity and halt cancer progression (Figure 1).

## 4. TXNIP and Inflammation and Immune Reaction

Inflammation, the immune system’s biological response, can be triggered by a variety of factors, including pathogens, damaged cells, and toxic substances. These elements can cause acute or chronic inflammation of tissues. As a result of cell damage, inflammatory cells become active, and several inflammatory signaling pathways are activated [6]. This section provides an overview of inflammatory reactions with TXNIP function. TXNIP is required for controlling NLRP inflammasome activation, according to new research. When exposed to oxidative stress, TXNIP is released and binds to the NLRP3 inflammasome, which activates inflammatory mediators such as IL-1B and IL-18, resulting in the induction of inflammatory immune responses [48]. The NLRP3 inflammasome is a cytoplasmic multiprotein complex that plays a role in the onset and progression of various diseases. It is made up of the effector protease precursor pro-cysteinyl aspartate specific proteinase-1 (pro-caspase-1), NLRs, and the apoptosis-associated speck-like protein with a caspase recruitment domain (CARD) (ASC) that implies the inflammation and immune response by activating inflammatory caspase-1 [49]. When NLRP3 is in the oligomerization stage, it interacts with ASC to recruit pro-caspase-1, which forms an oligomer and self-catalyzes the split into two subunits [7]. This interaction stimulates peril signals such as oxidative stress, potassium outflow, and crystal deposition [50,51,52]. Furthermore, the mature caspase 1 subunit directly mediated the immune response’s critical process promoted by IL-1β and IL-18 mediators [53].

## 5. Role of TXNIP in Cancer

Globally, cancer is one of the leading causes of death. By 2050, it will overtake all other diseases combined in terms of total mortality, especially among developing low- and middle-income countries [1]. Malignant tumor treatment is available in numerous forms, including surgical procedures, chemotherapy, immunotherapy, and radiotherapy. Unfortunately, despite significant advancements in managing malignant tumors, it is still difficult to determine the precise etiology of cancer, which has significantly hampered efforts to improve patient survival rates. It is currently believed that gene-targeted therapeutics approach may be key to alleviating the emergence and progression of human malignant tumors as an outcome of research aimed at the extensive understanding of genomic information and epigenetic changes. Therefore, the hunt for novel target genes is potentially significant for the formulation of cancer therapeutics.

TXNIP, with multiple functions, acts as a potent inhibitor of the thioredoxin system, where it prevents Trx (an antioxidant-defending protein system) from undergoing redox activity [54]. As a result, it leads to an exponential rise in reactive oxygen species that creates endoplasmic reticulum (ER) and cellular stress [55]. Therefore, TXNIP exhibits pro-oxidative stress, proinflammatory, and pro-apoptotic properties and thus has a huge role in cancer treatment. TXNIP performs significant functions in forming natural killer cells, controlling mitochondrial activity, stimulating cell-cycle arrest, energy metabolism, modulating glucose metabolism, inflammatory signal transduction, triggering apoptosis, and preventing proliferation and metastasis [56,57]. Therefore, focusing on suppressed TXNIP would be an effective therapeutic, and the cancers involved in its role are depicted in Figure 2.

### 5.1. TXNIP and Breast Cancer

Currently, the most prominent cancer, with approximately 2.3 million cases globally diagnosed yearly, with a lifetime risk of around 15%, is female breast cancer [58]. According to Ahmedin et al., 2022, new cases of breast cancer affected about 287,850 females and 2,710 males in the United States alone [59]. In human breast cancer, TXNIP has been shown to exhibit both prognostic and predictive value. Due to the strong correlation between elevated TXNIP mRNA expression and an improved prognosis, it has been suggested that TXNIP may be a suitable diagnostic biomarker for breast cancer in humans. Since TXNIP is a potent tumor inhibitor, it is frequently discovered that it is expressed at low concentrations across several tumor types [60]. TXNIP modifies the metabolic reprogramming in a breast cancer cell by influencing invasion, migration, and suppressing metastasis of cancer cells through the TXNIP-HIF1α-TWIST signaling axis [61]. Two important factors that trigger an epithelial-mesenchymal transition in tumor cells are twist-related protein (TWIST) and HIF-1α. In breast cancer, histone alteration may be a systematic mechanism of TXNIP transcriptional regulation, as evidenced by the rapid development of TXNIP mRNA in breast cancer cell lines which was seen when a histone deacetylase inhibitor was used to improve acetylation. Cancer cell development, progress, and proliferation eventually require glucose. The primary glucose metabolism regulator is TXNIP. TXNIP transcription initiation stimulates when the MondoA detects glucose-6 phosphate [62]. TXNIP inhibits glucose absorption both directly by adhering to GLUT1 and passively by alleviating the production of GLUT1 mRNA. Under the hypoxia state, cancer cells accelerate the overexpression of GLUT1 through TXNIP inhibition as the cancer cell depends on glycolysis during the hypoxia condition [63]. Compared to patients with other forms of breast cancer, individuals with triple-negative breast cancer (TNBC) have shorter life expectancies, a higher chance of recurrence rates, and an increased probability of developing metastases. In earlier research, it was shown that direct suppression of TXNIP by the transcription factor c-Myc promotes the metabolism of glucose in TNBC cells [64]. The primary reason for decreased TXNIP expression being one potential mechanism is that the MondoA and c-Myc fight for interactions with the domain of the TXNIP promoter that contains the E-box [65]. Therefore, the above research has proved that TXNIP could be a target for new treatment in TNBC patients.

### 5.2. TXNIP and Lung Cancer

With a global mortality rate of 21.6% and a prevalence of 12.4% in 2021, lung cancer will continue to be the top contributor to cancer-related deaths worldwide [59]. A significant subtype responsible for 82% of all lung cancer is non-small-cell lung cancer (NSCLC) [66]. Capable of interacting with tumor necrosis factor (TNF) receptor-related factor 6 (TRAF6), upregulating the tryptophan-aspartate repeat domain (WDR)5 expression, or suppressing Phosphoinositide 3-kinases (PI3Ks) and protein kinase B (AKT) signaling pathways, TXNIP can inhibit lung cancer cell multiplication and migration [67]. Therefore, the potential value of this pathway in cancer is acknowledged. Furthermore, it is essential to identify novel treatment pathway approaches for lung cancer patients. In a lung cancer cell, H4K5 acetylation and H3K4 trimethylation induced by sodium 4-phenylbutyrate (4PBA) were replaced by TXNIP-dependent sodium butyrate (NaBu) as they promote the WDR5 expression leading to the initiation of caspase 3/7 and induction of apoptosis [68]. TRAF6 integrates the NF-κB and RAS pathways to stimulate the progression of lung cancer. According to previous studies, NaBu-induced TXNIP can bind with TRAF6 via its Proline-Proline-x-Tyrosine (PPxY) motif and influence the synthesis and polyubiquitination of TXNIP, which in turn affects NSCLC development and migration [69]. It has recently been demonstrated that administering tyrosine kinase inhibitors (TKIs) to block PI3K/Akt signaling can significantly reduce the amount of GLUT1 that is bound to the cell membrane, despite increasing the synthesis of TXNIP in lung cancer tissues [70].

### 5.3. TXNIP and Prostate Cancer

The most common non-cutaneous category of cancer in males is prostate cancer (PCA). With an annual mortality rate of 350,000, it is the fifth most frequent reason for men to develop cancer globally. In the US, prostate cancer is the second most prevalent cause of death in men due to cancer [59,71]. Endocrine treatment is frequently used to treat severe PCA patients who are hormone sensitive, even though clinical cures are sometimes difficult to attain. Therefore, it is necessary to develop novel treatment targets for treating PCA. The protooncogene c-Myc induces metabolic reprogramming, which causes cells to become dependent on exogenous glucose and glutamine. For TXNIP expression to be downregulated and PCA cells to proliferate more quickly, c-Myc stimulates glutaminase 1 (GLS1) and inhibits the MondoA [65]. Ring Finger Protein (RNF)2 may be a new prognostic biomarker because it has been demonstrated that it is frequently overexpressed in several human malignancies and is highly correlated with a shorter overall expected lifespan. Furthermore, it has been shown that PCA has significantly increased RNF2 expression and that RNF2 inhibition can increase TXNIP expression by binding to the TXNIP promoter, which may result in PCA cell-cycle arrest, enhancing apoptosis and preventing tumor progression [72].

### 5.4. TXNIP and Cervical Cancer

In women, cervical cancer ranks fourth in terms of prevalence. The age-standardized occurrence of cervical cancer was 13.3% globally in 2020, while the death rate is projected at 7.3 per 100,000 women [73]. Ubiquitin-like containing PHD and RING finger domain 1 (UHRF1), which is essential in regulating DNA methylation, downregulates tumor suppressor genes when it is overexpressed, which contributes to oncogenesis in a range of cancer types, as well as in cervical cancer. According to recent research, human papillomavirus (HPV) E6/E7 oncoproteins promoted cell growth and substantially prevented cell death in cervical cancer by methylating the TXNIP promoter and down-regulating the expression of TXNIP. Thus, TXNIP is recommended as a potential therapeutic approach for cervical cancer [74]. Another study shows that in HeLa cells, TXNIP upregulation suppressed cell multiplication, migration, and invasion; however, in C-33A cells, TXNIP inhibition had the reverse impact. In addition, MondoA, instead of carbohydrate response element (ChRE)-binding protein (ChREBP), could be able to stimulate TXNIP expression in HeLa cells. Furthermore, MondoA overexpression suppressed HeLa cell motility, invasion, and proliferation via upregulating the TXNIP [75].

### 5.5. TXNIP and Liver Cancer

Epithelial or mesenchymal tissue present in the liver is where primary liver cancer develops [76]. Global Cancer Observatory (GLOBOCAN) estimated in 2020 that liver cancer accounted for over 906,000 new cases and 830,000 cancer-related mortality globally, making it the sixth most frequently diagnosed cancer [77]. Established medications or surgical resection cannot control the elevated occurrence of liver cancer metastasis and relapse [76]. Therefore, liver cancer has become a clinical challenge. TXNIP deficit is adequate for explaining hepatocellular carcinoma (HCC), which points to new pathways for how HCC develops. Recent studies showed that the exosomal miR-27a-3p produced by M2 macrophages enhanced the progression of HCC by downregulating TXNIP and stimulating the cancer stemness of HCC [78]. Another study revealed that TXNIP is downregulated by C-terminal-truncated hepatitis B virus X (Ct-HBx) and TXNIP has an essential role in Ct-HBx-induced growth of HCC. In HCC patients exhibiting Ct-HBx expression, TXNIP repression is commonly seen and significantly interlinked with a worse prognosis. Consequently, TXNIP expression suppressed the formation of tumors in the mouse model and decreased the metabolic reprogramming mediated by the Ct-HBx [79].

### 5.6. TXNIP and Leukemia

Acute myeloid leukemia (AML) is a condition identified by clonal growth that develops from progenitors of primitive hematopoietic stem cells with recurring genetic abnormalities [80]. In senior individuals, it has a very high death rate and a terrible prognosis. GLOBOCAN estimated the number of new cases to be around 474,519 (2.5%) and the number of deaths about 311,594 (3.1%) due to leukemia globally [77]. Epigenetic silencing is the primary cause of the reduced TXNIP expression in AML cells. According to a recent study, TXNIP upregulation inhibits the proliferation of AML cells with rearranged mixed-lineage leukemia. TXNIP promotes Beclin 1 expression to cause autophagy. TXNIP overexpression-induced autophagy promotes apoptotic induction, increasing sensitivity to the B-cell lymphoma (Bcl)2 and Bcl extra-large (XL) inhibitor, ABT263 [81]. Studies conducted by several researchers revealed that glucocorticoid response in children with acute lymphoblastic leukemia is controlled by the elevation of TXNIP expression.

### 5.7. TXNIP and Other Cancer

Currently, it is anticipated that TXNIP will be used as a molecular target for the treatment of various cancers. TXNIP could be regarded as a potential tumor suppressor gene since it is expressed at low concentrations in various malignancies. When TXNIP is overexpressed, it prevents the growth of cancer cells. Therefore, TXNIP has been hypothesized as a possible tumor suppressor in cancer due to its capacity to cause oxidative stress-mediated apoptosis in cancer cell lines and tumor tissues. The immune system’s ability to recognize and destroy tumor cells may be controlled by TXNIP’s immunological modulatory properties [5]. Multiple sources of evidence have demonstrated TXNIP’s irregular expression and prognostic role in many cancer types (Figure 2) and provided an understanding of the various molecular processes involved in developing malignant phenotypes.

## 6. TXNIP-Dependent Signaling Pathways in Cancer

TXNIP is known to play a crucial role in immune cells and immune responses during the progression of cancer. The studies that have examined cancer obstructions at various signaling pathways to date have found limited evidence of TXNIP’s contribution. In this decade, extensive studies have provided evidence supporting the role of TXNIP in connection with several mechanisms, including JAK-STAT, AMPK, PI3K, and other signaling pathways.

### 6.1. JAK/STAT Pathway

The transcription factor signal transducer and activator of transcription (STAT) are thought to be important in a variety of cellular processes, including the initiation of an immune response, cell survival, and cell proliferation [82]. Although there are seven highly conserved STAT isoforms in vertebrates, STATs 3 and 5 are most commonly seen in cancer through various pathways [83].

IL-6 cytokines mimic STAT3 by binding to and forming a complex with its receptor. Janus kinases (JAKs) are then phosphorylated and interact with the IL-6 receptor’s intracellular domains [84]. STAT3 is then activated by binding to the receptor’s phosphotyrosine. Furthermore, it dissociated from the receptor and formed dimers, which entered the nucleus and bound to DNA. As a result, STAT3 regulates genes involved in antiapoptotic cell cycle progression, and proliferation, with HIF-1α playing a significant role [85].

STAT3 hyperactivation was found in multiple myeloma, Hodgkin lymphoma, mantle cell lymphoma, and other cancers [86]. The Trx system has been shown to regulate the STAT3 pathway by stabilizing STAT3′s redox state [87]. In addition, researchers discovered that inhibiting Trx reductase can prevent STAT3-mediated transcription by keeping STAT3 from becoming oxidized. Trx reductase can also bind to STAT3 inhibitors. The Trx reductase inhibitor Auranofin, for example, has been shown to inhibit the STAT3 signaling pathway by preventing constitutive IL-6-induced signals of the JAK/STAT signaling pathway [88]. These findings unequivocally show that the Trx-dependent system is required for controlling and influencing STAT3 transcriptional functions. TXNIP, on the other hand, directly affects the Trx system. TXNIP binds inactive Trx, so there is no active Trx present when it is activated. Trx reductase is in charge of converting activated Trx to inactivated Trx; however, when TXNIP binds to inactivated Trx, Trx reductase is indeed not present [89] and suggests that TXINP has a direct impact on STAT3 regulation.

Protein tyrosine phosphatases (PTPs), which remove the phosphatase group from proteins, play an important role in JAK/STAT pathway regulation [90]. This PTPs dephosphorylation activity is redox regulated and loses functionality due to excess reactive oxygen species. It allows the JAK/STAT pathway to function effectively [91]. When TXNIP levels decrease, Trx levels rise and can lower ROS levels via antioxidants, which also reactivate PTPs, effectively terminating the cell signaling pathway [91]. Furthermore, Trx system inhibitors prevent PTPs from disrupting the signaling pathway. According to other studies investigating the role of PTPs proteins in many diseases, TXNIP activation may act as a tumor suppressor or an oncogene [92].

TXNIP acts as a tumor suppressor in glioblastoma by lowering STAT3 levels, which are increased by STAT3 inhibitors such as SS-4. Wang et al. demonstrated this in 2022. TXNIP expression increased in glioblastoma cells in response to SS-4 treatment, indicating that STAT3 tyrosine phosphorylation decreased TXNIP expression. Furthermore, TXNIP expression was significantly higher in STAT3^KO^ cells than in STAT3-rescued cells, which is consistent with the effect of SS-4 on STAT3-repressed genes in MT330 cells [93]. Overall, these findings suggest that TXNIP has the potential to regulate STAT3 signaling in immune cells involved in cancer development (Figure 3).

### 6.2. AMPK/mTOR Signaling Pathway

TXNIP is a tumor suppressor whose expression is reduced in various human cancers. TXNIP’s prognostic and predictive abilities in human breast cancer cases have been confirmed. Park et al., 2018, researched the clinical applicability and practical applications of TXNIP suppression in breast cancer. When compared to normal tissues, TXNIP expression was markedly downregulated in animal epithelium tumors and human breast cancer tissues. Tumors with high proliferative activity, such as those with high Kiel (Ki)-67 and low p27 affirmation, were found to have lower TXNIP protein levels [94].

Glucose is required for tumor cell growth and proliferation. TXNIP is an important regulator of glucose metabolism. When the transcription factor MondoA detects glucose-6 phosphate, the transcription of TXNIP, a potent negative regulator of glucose, is activated. TXNIP inhibits glucose uptake directly by binding to GLUT1 and indirectly by lowering GLUT1 mRNA levels and it was also discovered that inhibiting the upstream glycolytic enzyme glucose-6-phosphate isomerase significantly reduced GLUT1 expression after TXNIP induction, indicating that TXNIP regulates glucose metabolism [94,95].

The adenosine monophosphate (AMP)-activated protein kinase AMPK complex is a heterotrimeric complex composed of regulatory subunits with α β and γ subunits [96]. An increase in intracellular Ca2^+^/K^+^ levels activate calcium calmodulin-dependent kinase 2, activating AMPK signaling. K^+^ channels have been linked to the development of several types of malignant tumors [97]. Potassium channel subfamily K Member 3 (KCNK3), a genetic member of the two-pore domain potassium (K2P) channels, is still unknown in lung adenocarcinoma [98]. AMPK is activated in human prostate cancer samples, and blocking this pathway with a small interfering RNA or an AMPK inhibitor prevents cell growth [99,100]. AMPK can inhibit survival-enhancing downstream mammalian targets of rapamycin (mTOR) targets. mTOR signaling regulates a variety of critical cellular developments and processes. This pathway appears to be dysregulated in cancers and other disorders, and medications that inhibit it have been developed [101,102,103]. In contrast to the cellular effects of AMPK, mTOR activation increases anabolic processes such as protein and lipid synthesis [102]. Furthermore, TXNIP has recently been shown to be involved in AMPK signaling. According to one study, AMPK activation causes an increase in glucose uptake by inducing TXNIP degradation [104]. Similarly, numerous studies have shown that TXNIP inhibits mTOR signaling in prostate cancer and that mTOR signaling inhibition induces TXNIP expression [105,106]. On the other hand, it has been demonstrated mechanistically that when mTOR levels rise in cancer cells, it binds to MondoA directly in the cytoplasm and prevents the formation of the MondoA/Mlx complex which limits MondoA nuclear entry and TXNIP transcription and expression [107]. These findings collectively shed light on the critical role TXNIP plays in the AMPK/mTOR pathway in cancer cells. These findings suggest that TXNIP can potentially regulate AMPK/mTOR signaling in cancer (Figure 3).

### 6.3. PI3K/AKT Signaling Pathway

A common cause of constitutive activation of growth factor signaling, a factor in tumorigenesis, is genetic abnormalities in critical aspects of signaling pathways that rely on growth factors for signal transduction [108]. Cancer cells are known to preferentially produce ATP via anaerobic glycolysis in the presence of oxygen. The Warburg effect promotes tumorigenesis by directing metabolites to pathways where cancer can produce building blocks and antioxidants.

Phosphoinositide 3-kinases (PI3Ks) and protein kinase B (AKT) interact to promote cellular clearing and glucose metabolism. The molecular mechanism by which constitutive receptor tyrosine kinases (RTK) activation improves cellular glucose metabolism. In lung adenocarcinoma cell lines, inhibiting PI3K/mTOR metabolism also reduces glycolysis [109]. It is worth noting that inhibiting glycolysis resulted in a decrease in pentose phosphate and glycolytic pathway metabolites and an increase in GLUT1 in the cytoplasm under cancer conditions [110]. These findings imply that GLUT1 plasma membrane localization and aerobic glycolysis depend on the PI3K/mTOR pathway.

Furthermore, activation of the PI3K/Akt/mTOR pathway facilitates GLUT1 localization on the plasma membrane. TXNIP is the main factor that regulates GLUT1 plasma membrane localization [111,112]. It is well known that it inhibits glucose uptake into cells, negatively regulating glucose metabolism.

Through an increase in intracellular glucose-6-phosphate concentration, glucose activates MondoA/MLX [113]. Another study demonstrates that serum-induced TXNIP downregulation involves Ras-MAPK-dependent inhibition of TXNIP translation and PI3K/Akt-dependent inhibition of TXNIP transcription [114]. In addition, inhibition of constitutive PI3K/Akt signaling increases the expression of TXNIP in non-small-cell lung cancer, suggesting the potential significance of this mechanism in tumorigenesis. There have also been reports of a markedly higher incidence of chemically induced bladder cancer in TXNIP knockout mice [115]. Therefore, it is likely to be crucial for tumorigenesis when TXNIP expression is downregulated in cancers with constitutive PI3K/Akt/mTOR pathway activation.

TXNIP mediates the effect of constitutive growth factor signaling on cellular glucose metabolism, according to Hong et al. (2016). When PI3K/Akt signaling is activated, TXNIP transcription is downregulated. TXNIP expression, on the other hand, is increased in lung cancer cells by inhibiting activated PI3K/Akt signaling. TXNIP is also required for PI3K/Akt signaling to control GLUT1 plasma membrane localization, according to the researchers [116], which is also illustrated in Figure 3.

### 6.4. Other Pathways

Some of the major signaling pathways critical to cancer progression/impediments have already been discussed in detail above, to demonstrate that TXNIP plays a significant role in cancer impediments. Furthermore, TXNIP is implicated in many other pathways, including the HIF1α hypoxic pathway [117], the IL-dependent pathway [118], the B cell receptor (BCR) signaling pathway [119], and the NF-κB signaling pathway [120], all of which require empirical support to comprehend TXNIP’s role in cancer.

## 7. Therapeutic Targeting of TXNIP in Cancer

Many therapies have recently focused on immune cells and inflammatory mediators to treat and slow cancer spread. Current treatment regimens primarily target chemotherapy and proinflammatory mediators such as AMPK, HIF, PIK3/AKT, and other inflammatory precursors, but the recently discovered protein TXNIP has received less attention. With the aggressiveness of various immune cells in developing cancer and therapeutically targeting these molecules, it is critical to remember their potential and action during the disease process. TXNIP does this by regulating the Trx system and the activity of important immune cells. Cytokines, transcription factors, and inflammatory molecules are activated, resulting in cancer progression. Numerous preclinical studies have focused on upregulating TXNIP-driven molecules, which will soon lead to a much better way of treating various cancers (Table 1).

## 8. Future Perspective

TXNIP could be created as a biomarker for cancer with the help of all the available information. As one of the most frequently observed causes of human cancer, the TXNIP is now a strong candidate for therapeutic intervention. Under serum-free conditions, siRNA-mediated knockdown of TXNIP in PC-9 cells increased cellular glucose uptake and facilitated proliferation [116]. Therefore, TXNIP stimulation is required to control the progression of cancer cells. A novel and successful method for cancer treatment would be using small molecules or peptides as TXNIP activators. Immunogenic responses to TXNIP-related therapies and targeted signaling pathways are also two major issues that may be addressed to develop novel cancer treatments in the future.

On the surface, angiogenesis appears to be a major factor in cancer development, and TXNIP, a member of the arrestin family, is a potent inhibitor of tumor growth and angiogenesis as well as a suppressor of metastasis. Angiogenesis may be prevented by inhibiting Akt-mediated upregulation of vascular endothelial growth factors (VEGFs). As a hypothesis, TRX reductase, which activates TRX, may promote cell growth while preventing apoptosis. TRX also promotes the expression of HIF-1, VEGF, and angiogenesis. TXNIP prevents the biological effects of the active reduced form of TRX. As a result, researchers have attempted to bypass the lack of TXNIP by using TRX inhibitors or inhibiting TRX reductase. HIF-1 stimulates VEGF production, and HIF is inhibited by both [141]. A few other studies have attempted to clarify the relationship between TXNIP and angiogenesis in various cancers, however the picture remains unclear. Cancer-associated fibroblast-derived exosomes upregulated the miR-135-5p, which targets TXNIP in colorectal cancer and promotes cancer proliferation and angiogenesis by inhibiting TXNIP [142]. On the other hand, in renal cell carcinoma, TXNIP nuclear translocation with tortuous vessels caused tumor cell necrosis [143]. Notably, TXNIP knockdown in breast cancer cells promoted cell invasion and reversed the anti-angiogenic effects of p21WAF1 siRNA [144]. Sebastien et al., 2006, demonstrated that TXNIP is a prominent potential hypoxic marker in tumor angiogenesis. Since hypoxia is critical in cancer cell necrosis, TXNIP could serve as a prognostic tool as well as a potential anti-cancerous attractive therapeutic target in cancer [145]. Overall, the research on TXNIP and angiogenesis in various cancers is limited, highlighting the research gaps which must be extensively studied.

Furthermore, more research is needed to fully understand the TXNIP regulator’s precise mechanism in targeting apoptotic or repair pathways. TXNIP is being tested in clinical trials, and a better understanding of its role in conjunction with chemotherapeutic drugs is required to develop novel anticancer therapeutic approaches that specifically target cancers while also reversing chemoresistance. Because Trx system inhibition may reduce antioxidant production while also exerting anticancer activity, it is critical to fully understand these cross-talks as they will suggest a new line of investigation.

## 9. Conclusions

To summarize, there is ample evidence that TXNIP regulates immune responses and inflammation by maintaining cellular homeostasis. TXNIP is useful in preventing the progression of various cancers. TXNIP is now known to mediate several inflammatory signaling pathways and to be downregulated as cancer progresses. As a result, these findings could lay the groundwork for future and novel therapeutic interventions that target TXNIP activation. In cancer, preclinical studies are conducted on many approaches targeting TXNIP regulation. However, a mechanistic approach is required to understand the therapeutic targets for TXNIP signaling in cancers that use other pathways.

## Figures and Tables

**Figure 1 vaccines-10-01902-f001:**
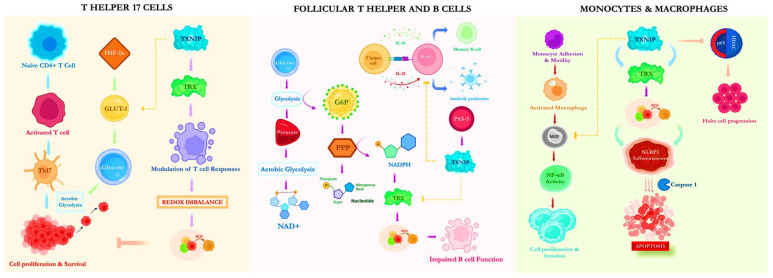
Mechanistic actions of TXNIP on different immune cells drive cancer impediments. Note: Th17: T-helper 17; HIF: Hypoxia-inducible factors; GLUT1: glucose transporter 1; TXNIP: Thioredoxin-interacting protein; Trx: Thioredoxin; ROS: Reactive oxygen species; NAD+: Nicotinamide adenine dinucleotide; G6P: Glucose 6 phosphate; PPP: Pentose phosphate pathway; IL: Interleukin; NADPH: Nicotinamide adenine dinucleotide phosphate; PAX: Paired box protein; MIF: Macrophage inhibiting factor; NF-κB: Nuclear factor kappa B; NLRP: NLR family pyrin domain containing; HDAC: histone deacetylases.

**Figure 2 vaccines-10-01902-f002:**
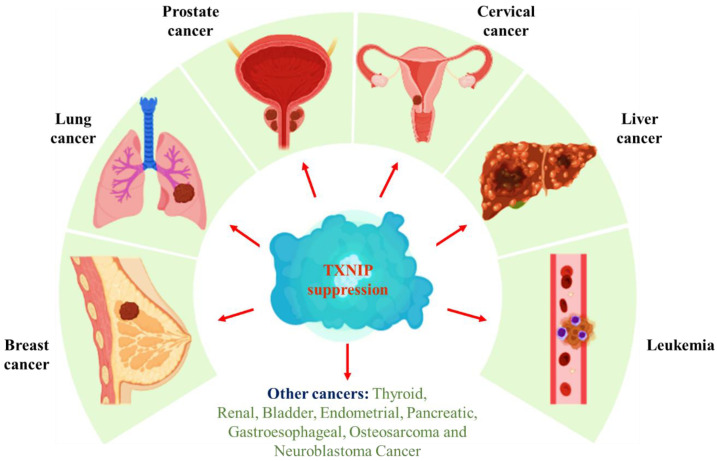
The systematic illustration of TXNIP suppression in cancer progression.

**Figure 3 vaccines-10-01902-f003:**
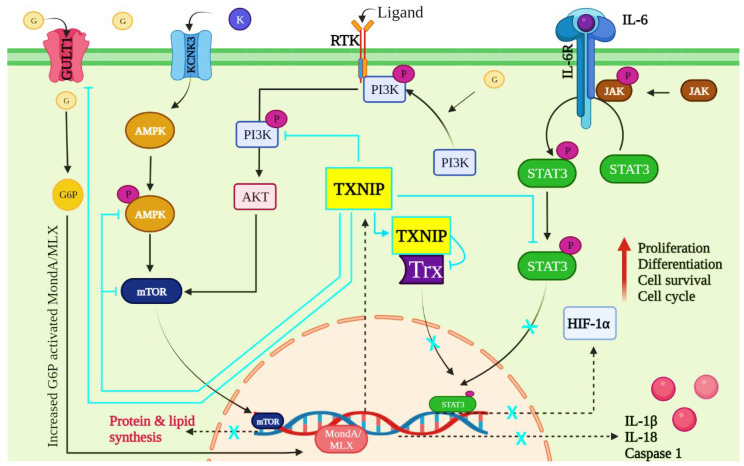
Schematic representation of TXNIP in mediating signaling pathway in cancer impediments. This picture represents how TXNIP is involved in cancer suppression via JAK/STAT, AMPK/mTOR, and PI3K/AKT signaling pathways. Note: Interleukins; JAK: Janus kinase; STAT: signal transducer and activator of transcription; HIF: Hypoxia-inducible factors; TXNIP: Thioredoxin-interacting protein; Trx: Thioredoxin; PI3K: Phosphatidylinositol 3 kinase; AKT: Protein Kinase B; RTK: Receptor tyrosine kinase; KCNK3: Potassium Channel, Subfamily K, Member 3; AMPK: AMP-activated protein kinase; mTOR: Mammalian target of rapamycin; GLUT1: Glucose transporter 1; G: Glucose; G6P: Glucose 6 phosphate; K: Potassium; p: Phosphorylation. The black line/arrow represents the regulating pathway, and the blue line represents the inhibitory role.

**Table 1 vaccines-10-01902-t001:** Novel therapies in preclinical studies for promoting TXNIP molecules in cancer suppression.

S.no	Compound Name	Compound Nature	Type of Cancer	Target/Mechanism	Model	Reference
1.	GW3965	LXR agonist	Pancreatic cancer	ATF4/TXNIP/REDD1/mTOR signaling	Cell lines MIA PaCa-2, BXPC3, hTERT-HPNE	[121]
2.	LHFPL3-AS2	Long non-coding RNA	Non-small cell Lung cancer	SFPQ-mediated transcriptional repression	BALB/c-nude mice	[122]
3.	KCNK3	Protein Coding gene	Lung adenocarcinoma	AMPK-TXNIP pathway	LUAD cell lines	[123]
4.	PLAG	Chemical synthesis	LLC1	TXNIP expression degrading A2BR	C57BL/6J mice LLC1-implanted TXNIP-KO	[124]
5.	3-BrPA	Organic compound	TNBC	c-Myc/TXNIP axis	TNBC (HCC1143) cell lines	[125]
6.	circ_SAR1A	Gene	Lung cancer	miR-21-5p/TXNIP axis	Lung cancer cell lines H1650, H1581, H460, H1299, and A549	[126]
7.	Citric acid	Organic compound	Ovarian cancer	CASP4/TXNIP-NLRP3-Gesdermin-d (GSDMD) pathway	Ovarian cancer cells	[127]
8.	DDIT4	Gene	Osteosarcoma	TXNIP/DDIT4/mTORC1 axis	Cell lines 143B, HOS, MG63, U2OS, and hFOB1	[128]
9.	N-WASP	Protein	squamous cell carcinoma	N-WASP-ERK2-FOXO1-TXNIP pathway	Cell line HSC-5	[129]
10.	WT1	Protein	Renal cell carcinoma	WT1/IL-24/TXNIP axis	Cell line A498	[130]
11.	circDCUN1D4	Gene	Lung adenocarcinoma	circDCUN1D4/HuR/TXNIP RNA-protein ternary complex	Cell lines HBE, A549, H1299, H1975, PC-9, and SPCA-1	[131]
12.	MAGI2-AS3	Long non-coding RNA	Non-small cell Lung cancer	miR-629-5p/TXNIP axis	Cell lines Beas-2B, A549, and H1299	[132]
13.	Silibinin	flavonoid	TNBC	EGFR-MYC-TXNIP axis	Cell line MDA-MB-231, MDA-MB-468, BT549, BT474, and SKBR3	[133]
14.	Ceramide	Lipid molecules	Non-small-cell lung cancer	Txnip/Trx1 complex	Cell lines A549 and PC9	[134]
15.	HDAC10	Enzyme	Cervical cancer	microRNA-223/TXNIP/Wnt/β-catenin pathway	CC cells C33A, MS751, HeLa, SiHa, and End1/E6E7	[135]
16.	cRAPGEF5	Circular RNA	Renal cell carcinoma	miR-27a-3p/TXNIP pathway	Cell lines 769-P, Caki-1, OSRC-2, and 786-O	[136]
17.	Amikacin	Antibiotic	Breast Cancer	TXNIP	cell line MDA-MB-231	[137]
18.	Vitamin D3	Fat-soluble vitamin	Endometrial cancer	TXNIP	Human endometrial cancer cell line	[138]
19.	cytarabine	Nucleoside analog	Acute myeloid leukemia	TXNIP-ASK1-JNK1 axis	Cell lines, HL60 and U937	[139]
20.	D-allose + docetaxel	Sugar + semi-synthetic drug	Head and neck cancer	TXNIP	Cell line HSC3	[140]

Note: ATF4: Activating transcription factor 4; REDD1: regulated in development and DNA damage responses 1; mTOR: mammalian target of rapamycin; LHFPL3-AS2: LHFPL3 Antisense RNA 2; SFPQ: Splicing factor proline and glutamine-rich; LUAD: lung adenocarcinoma; KCNK3: potassium two-pore domain channel subfamily K member 3; PLAG: palmitoyl-2-linoleoyl-3-acetyl-rac-glycerol; AMPK: AMP-activated protein kinase; LLC1: Lewis lung carcinoma; TXNIP-KO: TXNIP-knockout; A2BR: adenosine 2B receptor; 3-BrPA: 3-bromopyruvic acid; TNBC: triple-negative breast cancer; DDIT4: DNA damage-inducible transcript 4; N-WASP: neural Wiskott–Aldrich syndrome protein; ERK2: extracellular signal-regulated kinase 2; FOXO1: forkhead box protein O1; WTI1: Wilms’ tumor gene 1; IL-24: Interleukin 24; circDCUN1D4: circ-defective in cullin neddylation 1 domain containing 4; HuR: human antigen R; MAGI2-AS3: MAGI2 antisense RNA 3; EGFR: epidermal growth factor receptor; HDAC10: histone deacetylase 10; Wnt: wingless-related integration site; ASK1: apoptosis signal-regulating kinase 1; JNK1: c-Jun N-terminal protein kinase 1.

## Data Availability

Data are contained within the article.

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
