# Peer review of "Immunomodulatory Role of Thioredoxin Interacting Protein in Cancer’s Impediments: Current Understanding and Therapeutic Implications"

_vaccines, 2022, doi:10.3390/vaccines10111902_

Round 1

Reviewer 1 Report

This review article provides a current and well-articulated understanding of the role thioredoxine plays in cancer development, and provides useful therapeutic suggestions and potential future guidelines in the oncology field. The manuscript provides an indepth assessment of the role of thioredoxine in regulating downstream signlaing pathways as well as inflammation and immune reactions, and highlights the role of the protein in the most common forms of cancers in both men an women. 

MInor comments:
a few minor typos should be amended. Also, at lines 259, the number of female affected by breat cancer should be amended for clarification

Author Response

Reviewer 1

Minor comments:
a few minor typos should be amended. Also, at lines 259, the number of females affected by breast cancer should be amended for clarification

Response: As per reviewer suggestion typos are rectified and line 259 has been clarified. The clarified statement is “According to Ahmedin et, al., 2022, new cases of breast cancer affected about 2,87,850 females and 2,710 males in the United States alone”

Reviewer 2 Report

The manuscript by Karrurajan et al. aimed to update the knowledge about the role of thioredoxin interacting protein (TXNIP) in cancer. This manuscript is well-organized; however, there are several points needing to be addressed.

1. The resolution of figure 1 could be improved.

2. What’s the mechanism for the activation or expression of TXNIP during the cancer development? Authors should discuss this point in revised manuscript.

3. Angiogenesis is a key step for tumorigenesis, what’s the role and regulatory mechanism of TXNIP in the process of angiogenesis? Authors should discuss this point in revised manuscript. 

Author Response

Reviewer 2

  1. The resolution of figure 1 could be improved.

Response: As per reviewer suggestions resolutions of the figure 1 improved

  1. What’s the mechanism for the activation or expression of TXNIP during the cancer development?

Response: Although TXNIP expression is substantially reduced as cancer progresses, as outlined in this revised manuscript, Table 1 shows that TXNIP levels are activated by a number of approaches that may act as a formidable competitor in regulating cell metabolism and proliferation.

  1. Angiogenesis is a key step for tumorigenesis, what’s the role of TXNIP in angiogenesis?

Response: Yes, angiogenesis is a key step for tumorigenesis. Since researches are limited in relationship with TXNIP and angiogenesis, the related information is incorporated in future perspective section

Round 2

Reviewer 2 Report

Authors could discuss, at least, the issue about TXNIP and angiogensis in cancer. The reviewer has listed several references as below:

1. Cancer-associated fibroblasts-derived exosomes upregulate microRNA-135b-5p to promote colorectal cancer cell growth and angiogenesis by inhibiting thioredoxin-interacting protein. Cell Signal. 2021;84:110029.

2. Expression of TXNIP is associated with angiogenesis and postoperative relapse of conventional renal cell carcinoma. Sci Rep. 2021;11(1):17200.

3. The cyclin-dependent kinase inhibitor, p21(WAF1), promotes angiogenesis by repressing gene transcription of thioredoxin-binding protein 2 in cancer cells. Carcinogenesis. 2009;30(11):1865-71.

 4. Characterization of the expression of the hypoxia-induced genes neuritin, TXNIP and IGFBP3 in cancer. FEBS Lett. 2006;580(14):3395-400.

Author Response

Reviewer

Comments and Suggestions for Authors

Authors could discuss, at least, the issue about TXNIP and angiogenesis in cancer. The reviewer has listed several references as below:

  1. Cancer-associated fibroblasts-derived exosomes upregulate microRNA-135b-5p to promote colorectal cancer cell growth and angiogenesis by inhibiting thioredoxin-interacting protein. Cell Signal. 2021;84:110029.
  2. Expression of TXNIP is associated with angiogenesis and postoperative relapse of conventional renal cell carcinoma. Sci Rep. 2021;11(1):17200.
  3. The cyclin-dependent kinase inhibitor, p21(WAF1), promotes angiogenesis by repressing gene transcription of thioredoxin-binding protein 2 in cancer cells. Carcinogenesis. 2009;30(11):1865-71.
  4. Characterization of the expression of the hypoxia-induced genes neuritin, TXNIP and IGFBP3 in cancer. FEBS Lett. 2006;580(14):3395-400.

Response: As per the reviewer’s suggestion “the issue about TXNIP and angiogenesis in cancer” has been discussed with given references in appropriate place and content. Same has been highlighted